# The Three Pillars of Glioblastoma: A Systematic Review and Novel Analysis of Multi-Omics and Clinical Data

**DOI:** 10.3390/cells13211754

**Published:** 2024-10-23

**Authors:** Ciro De Luca, Assunta Virtuoso, Michele Papa, Giovanni Cirillo, Giuseppe La Rocca, Sergio Corvino, Manlio Barbarisi, Roberto Altieri

**Affiliations:** 1Laboratory of Neuronal Networks Morphology and System Biology, Department of Mental and Physical Health and Preventive Medicine, University of Campania “Luigi Vanvitelli”, 80138 Naples, Italy; assunta.virtuoso@unicampania.it (A.V.); michele.papa@unicampania.it (M.P.); giovanni.cirillo@unicampania.it (G.C.); 2ISBE Italy, SYSBIO Centre of Systems Biology, 20126 Milan, Italy; 3Department of Neurosurgery, Fondazione Policlinico Universitario “A. Gemelli” IRCCS, Catholic University of Rome School of Medicine, 00153 Rome, Italy; giuseppe.larocca@policlinicogemelli.it; 4Department of Neurosciences and Reproductive and Odontostomatological Sciences, Neurosurgical Clinic, University “Federico II” of Naples, 80131 Naples, Italy; sercorvino@gmail.com; 5Multidisciplinary Department of Medical-Surgical and Dental Specialties, University of Campania “Luigi Vanvitelli”, 80131 Naples, Italyroberto.altieri@unicampania.it (R.A.)

**Keywords:** glioblastoma, neurovascular unit, immune system, cancer stem cells, multi-omics

## Abstract

Glioblastoma is the most fatal and common malignant brain tumor, excluding metastasis and with a median survival of approximately one year. While solid tumors benefit from newly approved drugs, immunotherapy, and prevention, none of these scenarios are opening for glioblastoma. The key to unlocking the peculiar features of glioblastoma is observing its molecular and anatomical features tightly entangled with the host’s central nervous system (CNS). In June 2024, we searched the PUBMED electronic database. Data collection and analysis were conducted independently by two reviewers. Results: A total of 215 articles were identified, and 192 were excluded based on inclusion and exclusion criteria. The remaining 23 were used for collecting divergent molecular pathways and anatomical features of glioblastoma. The analysis of the selected papers revealed a multifaced tumor with extreme variability and cellular reprogramming that are observable within the same patient. All the variability of glioblastoma could be clustered into three pillars to dissect the physiology of the tumor: 1. necrotic core; 2. vascular proliferation; 3. CNS infiltration. These three pillars support glioblastoma survival, with a pivotal role of the neurovascular unit, as supported by the most recent paper published by experts in the field.

## 1. Introduction

Glioblastoma is the most frequent and fatal primary malignant brain tumor, with a median survival for patients receiving the current evidence-based treatment between 12 and 15 months [1,2]. During tumor progression, there is selective pressure for new mutations that arise within the tumoral milieu and further specialize cellular and extracellular components of the developing mass, with genomic and environmental heterogeneity. These changes account for drug resistance and the short-term efficacy of state-of-the-art protocols [3].

Therefore, the current understanding of glioblastoma as an uncontrolled proliferating tissue can be discarded. Glioblastoma develops within the central nervous system (CNS) and perfectly adapts to its features, mimicking CNS components’ anatomy and metabolism. Glioblastoma is a system fully equipped to hack the CNS and therefore cannot metastasize elsewhere. 

The best approaches to understanding the features of glioblastoma are the spatial and multi-omics data analyses that have flourished in recent years [1,4,5,6]. The major drawback of the big data generated is the interpretation of the experimental outcome and the failure to translate it into clinical protocols. Moreover, the CNS anatomy seems to be tightly intertwined with glioblastoma. The Cancer Genome Atlas (TCGA) and the REpository for Molecular BRAin Neoplasia DaTa (REMBRANDT) are two of the main multidisciplinary efforts that seek to overcome the difficulty of collecting and understanding multi-omics data within the human brain and tumors [7,8]. Cutting-edge technologies should be linked to standard imaging for translating the research results into clinical settings where MRI protocols are used to diagnose and follow up patients with glioblastoma [9].

In 2021, the WHO classification [10,11] was updated to express better brain tumors’ molecular and prognostic features. For the first time, adult-diffuse gliomas were separated from pediatric ones. However, pediatric-type tumors may rarely occur in adults and vice versa [9]. The role of molecular biomarkers for classifying gliomas reflects all the efforts for evidence-based stratification in diffuse gliomas [10]. Isocitrate dehydrogenase (IDH) mutational testing should be performed on all diffuse gliomas, and other molecular features are essential for establishing the grade. Nongenetic biomarkers, such as O(6)-methylguanine-DNA-methyltransferase (MGMT) promoter methylation status, were not included in the 2021 WHO classification but are considered strong prognostic predictors [12] and should be performed on glioblastoma, IDH-wildtype. 

Glioblastoma, IDH-wildtype is the only glioblastoma in the new classification, while the previous “glioblastoma, IDH mutant” is now classified as “astrocytoma, IDH-mutant” (covering WHO grades 2–4). Glioblastoma, IDH-wildtype (from this point forward, simply glioblastoma) is an adult-type brain tumor within the category of diffuse gliomas with three subtypes: giant cell glioblastoma, gliosarcoma, and epithelioid glioblastoma [10]. Additional genetic hallmarks of glioblastoma are telomerase reverse transcriptase (TERT) promoter mutation, epidermal growth factor receptor (EGFR) amplification, and/or +7/−10 chromosomes copy-number alterations [11].

Intratumoral heterogeneity is mainly established through these genetic and phenotypic biomarkers. Microenvironmental heterogeneity, on the other hand, is characterized by the interaction with other nonmalignant cell types [13]. TCGA and REMBRANDT helped to shed light on the heterogeneity and the pathogenesis of different types of brain cancers, including gliomas and glioblastoma. However, both failed to discover what seems crucial: the intratumor cellular and environmental heterogeneity. Single-cell studies have furnished a spatially defined signature that can occur within the glioblastoma mass [14,15,16,17,18]. Although promising, these protocols underestimate the systems biology, focusing on cellular divergent molecular pathways. Accounting for single-cell divergence is important; however, we propose to include a clustering system for the anatomical variability of different types of glioblastoma through three biological pillars: 1. necrotic core, 2. vascular proliferation, 3. CNS infiltration. These features are shared among all forms of glioblastoma and must subtend the convergent path that has not been underlined yet. The important and noble efforts to unravel glioblastoma molecular differences (intra- and extratumor) can lead to tailored treatments. However, a successful therapy needs to interfere with these three common pillars at the same time. The three pillars are fixed around the central nodes of immune escape and metabolism (Figure 1). Immunity and metabolic rewiring are orchestrated by the neurovascular unit (NVU) and the cancer stem cells (CSCs). The NVU has remarkable importance in different pathologies, as the evidence enlightens, and the development of the glioblastoma within the CNS cannot proceed without affecting it [19,20]. Furthermore, the NVU has peculiar anatomical differences in subventricular and hippocampal regions, which are the most divergent anatomical phenotypes of glioblastoma. The CSCs instead might be the origin of gliomagenesis due to their unique biological features, particularly their capability for self-renewal, differentiation, and migratory potential. In a previous paper, we were the first to demonstrate in vivo their presence in the central core and peritumoral areas at similar concentrations [21]. The two brain regions that manifest the highest demonstrated concentration of stem cells are the subventricular zone (SVZ) and the hippocampus [19,22]. This speculation agrees with the anatomical and morphological differences supported by multi-omics studies. Different tumors, with specific features, could arise from definite spatial locations [23]. Many researchers have emphasized the significant role of the tumor microenvironment and the interaction between microglia/macrophages and the extracellular matrix (ECM). Indeed, the connection between microglia and tumor cells was first proposed by Sir Wilder Penfield in 1925 [24], and considering recent discoveries can be included in the proposed model of NVU–CSC interplay and the three glioblastoma pillars.

From a radiological perspective, we proposed a model for glioblastoma that describes its growth rate using real-life clinical-setting radiological markers [25]. These findings indicate that glioblastoma has a high metabolic activity, particularly in a moving front of rapid proliferation situated within the enhancing nodule detected using the T1 sequence of MRI following gadolinium injection. In contrast, the peripheral zone exhibits lower metabolic activity, which increases as the central core becomes necrotic. This phenomenon might be explained by considering the natural progression and maturation of glioblastoma in the context of a Gompertzian curve [26].

However, these multi-omics, clinical, and radiological data remain disparate elements of an incomplete puzzle. We propose to review the literature systematically and convey the obtained results to shape a theory that can give another perspective on the anatomy of glioblastoma. We consider human multi-omics research and real-life clinical settings to highlight the spatial location and biological signatures of glioblastoma within the proposed three pillars of core necrosis, vascular proliferation, and CNS infiltration (Figure 1). The high mitotic activity of glioblastoma cells is a key feature; however, it can be a common element of the three pillars, accounting for clonal selection in the necrotic core, metabolic demand for microvascular proliferation, and infiltrative potential [27,28,29], so it will not be treated as a separate pillar.

## 2. Methods

In June 2024, the PUBMED electronic database was searched, and the following terms were applied: (“spatial” or “atlas”) and “neurosurgery” and “human” and (“glioblastoma” or “GBM”)) not “review”. The results were stored and managed with the ZOTERO reference manager version 6.0.36 (Center for History and New Media, George Mason University, Virginia, VA, USA). All papers written in languages other than English were excluded. Time restrictions were applied considering articles published after 2021, after the most recent World Health Organization classification of glioblastoma. The original articles published after this date contain up-to-date molecular features that can be analyzed considering the novel classification. However, the discussion will include papers based on previous classification criteria to enrich the analysis. This bias cannot be avoided considering the recent update in WHO classification. All processes are defined in the PRISMA [30] flowsheet (Figure 2). We did not register the review protocol.

The inclusion criteria are as follows:

Multi-omics approach;Retained spatial information of the tumor;Human-derived data;Novel experimental data.

The exclusion criteria are as follows:Articles without human patients;No spatial information;No full text available;Full text not in English.

## 3. Results

A total of 215 articles were identified using the search algorithm on PUBMED. We used the search term to include papers specifically aimed towards research articles of human-derived data containing spatial information. One article was not in English and was not screened. Two independent reviewers reviewed the titles and abstracts of all the articles and sought retrieval of 113 out of 215 to include all the articles that could adhere to the inclusion criteria. We excluded 30 papers for lack of human-derived novel data and 60 for insufficient spatial definition. The remaining 23 were considered for their clinical and basic research approaches for writing this review (Figure 2).

### 3.1. First Pillar: Necrotic Core

The core of glioblastoma is the initial location of the differentiation of CSCs. CSCs play a significant role in tumorigenesis and chemoradiotherapy resistance, being considered among the candidates for immunotherapy failure in this cancer. Treatment failure in glioblastoma is attributed to intratumoral heterogeneity providing clonal selection for-resistant CSCs [31]. Intratumoral heterogeneity can occur with genetic mutations, differences in chromatin accessibility, or transcriptional regulation. Moreover, the nonmalignant cells in the microenvironment (neurons, macro- and microglia, and immune populations) functionally interact with glioblastoma [31]. Intratumor heterogeneity is the intrinsic integration of diverse stimuli from genetic, phenotypic, and microenvironmental heterogeneity and is particularly important in glioblastoma [13].

A novel spatially related stemness-based classification of glioblastoma was proposed outlining different populations of CSCs using a one-class logistic regression algorithm to calculate an mRNA-based stemness index of 518 patients from TCGA database (transcriptomics data of glioblastoma and pluripotent stem cells) [32]. Based on the signature of these cells, the classification generated two subtypes via consensus clustering. Patients in stemness subtype I (SSI) showed a better overall survival, paralleled by a reduced progression-free survival compared to SSII. SSI had an increased somatic mutation load and burden of copy number alterations. Distinct tumor microenvironmental changes were also apparent after the clustering. The relevance of this transcriptomic and spatial classification was further confirmed by the higher responsiveness to immunotherapy with anti-PD1 treatment and higher resistance to temozolomide of the SSI group. Multiple machine learning algorithms were used to characterize a predictor, based on seven genes validated in independent glioblastoma cohorts. The preliminary stemness classification based on TCGA could pave the way for selective treatment, with a central role of the immune system [32]. 

Indeed, the core of the glioblastoma showed differences in IDH-wildtype versus astrocytoma, IDH-mutant grade 4, with a reduced proportion of cytotoxic/immature NK cells and a higher frequency of macrophages but not microglia, suggesting an increased peripheral margination and invasion of the tumoral core by macrophages proportional to disease severity [33]. Recently, long-term survival (more than three years) in glioblastoma was not correlated to MGMT methylation status, usually considered a strong prognostic indicator, but to an increased endothelial invasion of the core [34]. Although unexpected, the observed phenomenon could be related to higher efficiency in chemotherapy delivery. Alternatively, the vascular niche, as detailed in the second pillar, can modify the immune reactivity of some patients. The presence of CD8- and CD4-positive or -negative T-cells but non-T-reg in the microenvironment was associated with increased survival [34,35]. A sexual difference was observed corresponding to higher frequencies of peripherally derived macrophages and an increased concentration of endothelial cells in male patients, suggesting a differential and personalized therapeutical approach according to gender. 

Another approach to define and classify specific spatiotemporal alterations of the glioblastoma core led to five distinct transcriptional programs with shared alterations [36]. The description considers the copy number alteration profile with the regional pattern and the transcriptional signature. The comparison allowed authors to confront the results with the WHO classification and compare simultaneously cellular and histological patterns of the glioblastoma [10,36]. This different approach marked the dynamic variations of cellular states within the tumor and its microenvironment, suggesting peculiar metabolic alterations. Hypoxia emerged as the principal cause of metabolic impairment, followed by proliferation block in the S phase and the accumulation of genomic alterations as copy number alterations [36]. TCGA data validated the hypothesis of regional hypoxia paired with genomic instability [37,38]. Considering the long-term survival observed in highly vascularized glioblastoma cores, this evidence is corroborated [34]. The hypothetical model lacks clinical mechanistic validation; however, it was investigated through human and rodent neocortical tissue sections inoculated with patient-derived primary glioma stem cells, preserving all the spatial, environmental, and cellular characteristics aside from the neurovascular unit. The age of the donor and the inflammatory response were critically related to tumor differentiation [36]. The in vitro result can be matched with the increased transformation potential and the incidence of glioblastoma in aged individuals [39]. 

The correlation between aging, inflammatory status, and glioblastoma was suggested, although without a causal link. Other neurological inflammatory or degenerative pathologies, (e.g., Alzheimer’s disease and multiple sclerosis) prompt a general inflammatory environment and transformation of glial cells [40] directly or by damage to the neurovascular unit [41]. 

The elucidation of the differential regional transcriptional profiles of glioblastoma with metabolic, tumor–host cellular interactions, and microenvironmental variations has improved our knowledge about the first pillar of glioblastoma and allowed the interpretation of the connections with the other two pillars. The inter-patient heterogeneity with hypoxia as a major factor for genomic and transcriptional abnormalities is one of the most promising perspectives for personalized therapies avoiding drug resistance and early relapse. The microglia/macrophagic invasion has been proposed as a regulating mechanism for the proneural–mesenchymal phenotype switch of glioblastoma worsening the prognosis [42]. Iba-1 levels are indeed more abundant in mesenchymal glioblastoma. The tumor stroma is filled with tightly packed macrophages/microglia whose role in contributing to a poor prognosis needs to be elucidated. The microenvironment of the stroma or the nest of the glioblastoma could regulate the function of these cells, but there are no conclusive data [43].

The microglia/macrophage lineages are not a homogenous population. Discovering the molecular and functional heterogeneity of these cells in human glioblastoma is still a challenge, having surpassed the in vitro-defined phenotypes (M1/M2) and including different roles for the embryonic-yolk-sack-derived microglia (CNS resident) and the bone-marrow-specialized monocytes/macrophages (infiltrating) [6,44]. Overall, myeloid cells associated with glioblastoma secrete signal molecules (cytokines) and metabolites inhibiting lymphocytes. While the stroma and nest of a tumor are well defined on other solid masses and fibroblasts are the main source for the stroma of gastric or breast cancers, for glioblastoma, the composition and the boundaries between the nest and stroma are less defined. The high expression of MHCII seems to help distinguish the stromal areas in human glioblastoma. Moreover, levels of MHCII increase with the proliferation index Ki67, which corroborates the idea that a low tumor–stroma ratio indicates more proliferative tumors and aggressiveness [45]. 

A definite pathological relevance of stem cells other than CSCs has also been proposed: mesenchymal stem cells (MSCs), neural precursor cells (NPCs), and oligodendrocyte precursor cells (OPCs). However, a definite role for each cytotype in the tumor core has not been validated. MSCs revealed tumor-cradling and antitumoral potential according to their origin or localization [22,46,47]. NPCs, more robustly, seem to impair glioblastoma progression by the active release of inhibitors [46]. The relevance of the NPCs’ decline with aging could be paired with glioblastoma incidence. NPC-mediated suppression of glioblastoma should therefore be confined to the young brain. OPCs’ role has also been postulated for their contribution to the perivascular cells of glioblastoma, but there is scarce evidence of their contribution to glioblastoma progression [47].

### 3.2. Second Pillar: Vascular Proliferation

At least part of the poor prognosis for glioblastoma patients is given by the scarcity of therapeutic options available following surgery and radiotherapy. All the efforts made by decades of modern and valuable research have resulted in the FDA approving only four drugs for glioblastoma treatment: bevacizumab, carmustine, lomustine, and temozolomide (https://www.cancer.gov/about-cancer/treatment/drugs/cancer-type/ accessed on 11 July 2024).

These therapies, and other antitumor compounds tested, including low-molecular-weight compounds, must endeavor against the blood–brain interface, characterized by a well-defined unit from the vascular lumen to the synaptic cleft, the NVU, granting the homeostasis of the CNS. However, failure to cross the first space of the unit, the so-called blood–brain barrier (BBB), is one of the unique features of the brain that has led to negative outcomes of tested drugs.

#### 3.2.1. Endothelial Cells and Pericytes

Endothelial cells are the first element facing compounds in the bloodstream and one of the pillars of glioblastoma. The endothelium of the brain vessels, excluding the circumventricular organs [48] (CVOs: area postrema, median eminence, neurohypophysis, organum vasculosum of the lamina terminalis, pineal gland, subcommissural organ, and subfornical organ), are essentially provided with a distinct set of tight junctions (TJs) that allow a selective and cell-mediated transport of the substances, limiting the passive diffusion to small and lipophilic molecules that are not actively expelled. Indeed, the endothelial barrier is furnished with a prompt set of tunable transporters that bind and limit or favor the influx/efflux of molecules toward/from the brain [48,49]. Endothelial cells regulate the passage of water-soluble molecules independently of their molecular weight through pinocytosis and vesicular transcytosis. All these characteristics are important for CNS homeostasis, and data from cellular transcription isolated from abnormal vessels in glioblastoma show a distinct gene signature and vascular proliferation as key features among all glioblastoma phenotypes [50,51,52]. 

Nonetheless, one of the most recent drugs approved for glioblastoma, bevacizumab, is a monoclonal antibody blocking the vascular endothelial growth factor (VEGF), which did not improve overall survival in unselected patients and led to a reduction in temozolomide delivery, altering the vascular remodeling [53]. This failure is representative of why all three proposed pillars should be addressed simultaneously in designing a targeted treatment. Studying brain endothelial cells at the single-cell level, with RNA sequencing (scRNA-seq) of the human glioblastoma tumor core and peritumoral tissue, showed characteristic heterogenic gene expression [54]. Five endothelial cell signatures have been clustered based on this study, associated with both peculiar anatomical localizations and distinct molecular assets of glioblastoma. Noteworthy, the endothelial transporters usually expressed in the physiological BBB were found mostly retained in diverse glioblastoma signatures. Their expression was heterogeneous rather than downregulated or absent. These data further corroborate the idea of glioblastoma maintaining the physiological systems’ biology even among the most abnormal vasculature, to be cradled inside the CNS. The first scRNA-seq molecular atlas of the human BBB in glioblastoma provided this evidence [54].

Although the endothelial transporters are preserved, forming new capillaries sprouting from the existing arterioles requires extensive modifications. This allows glioblastoma to generally be a highly vascularized tumor with a growth rate consistently limited by the metabolic supply [55]. For instance, endothelial cells modify cell–cell contact, the metabolic and proliferation rates, and differentiation. External stimuli from the peritumoral and the tumoral environment, particularly those mediated by surface receptors, are responsible for vascular reorganization [56]. Pericytes mediate neo-angiogenesis in the glioblastoma mass, contacting both tumoral perivascular cells and brain endothelium through Semaphorin-4D (SEMA4D) and focal adhesion kinase (FAK). The SEMA4D protein or Cluster of Differentiation 100 (CD100) is a well-known cell-guidance factor that seems crucial for pericyte migration and neo-angiogenesis in combination with high levels of FAK expression. SEMA4D and FAK are associated with poorer prognosis in high-grade gliomas [57].

Aside from these single specific markers of neo-angiogenesis, normal brain tissue and glioblastoma endothelial cells were studied using scRNA-seq and bulk RNA-seq data [58]. That study described a model for glioblastoma patients’ prognosis that was successfully validated using TCGA and Chinese Glioma Genome Atlas (CGGA) cohorts of patients. The characterization of endothelial cells into clusters was able to distinguish patients into high- and low-risk groups. The study also showed that higher calculated risk was associated with poorer patient prognosis, lower frequency of isocitrate dehydrogenase (IDH)1/2 mutations, and Programmed Death Ligand 1 (PD-L1) upregulation. The model showed the importance of the tumor microenvironment (TME) in shaping the risk score with different prognoses [59]. 

Spatial RNA expression enlightened the endothelial genes pivotal for sustaining this pillar: TUBA1C, RPS4X, KDELR2, and SLC40A1. TUBA1C is an isoform of the alpha-microtubule cytoskeletal protein, paramount for cell overall motility, from vesicular trafficking to cell migration, phase transition, and proliferation [60]. Indeed, a knockdown of TUBA1C showed inhibited migration, a phase arrest in G2/M, inhibited proliferation, and enhanced apoptosis [61]. 

RPS4X (ribosomal protein S4 X-linked) is an oncogene described to increase cisplatin resistance and associated with poor survival and disease progression in ovarian cancer [62]. Although the role of RPS4X in glioblastoma has not been reported yet, it should act through the mTOR signaling pathway [63]. The KDELR2 (KDEL receptor2) can consistently promote the stability of hypoxia-inducible factor 1a (HIF1a) through mTOR and prompt glioblastoma angiogenesis [64]. KDELR2 knockdown induces cell cycle arrest (G1 phase) and apoptosis, reducing cell viability. Lastly, SLC40A1 (Solute carrier family 40, member 1) is already described in ovarian cancer and multiple myeloma, where it inhibits cell growth and reduces chemotherapy resistance. An in silico study suggested the role of SLC40A1 as a ferroptosis suppressor, associated with immunosuppression in gliomas [65]. Classical gene mutations (TP53 and PTEN) were not correlated with differences in overall survival in a recent model based on these factors. However, IDH mutations were present in the low-risk population, validating the reliability of the model and the pivotal importance of the first front of the vascular interface [66].

Complex in vitro models are being developed to test the blood–brain interface in reproducible and consistent scenarios. A perfused tumor spheroid with a dense tumor core and perivascular space invasion was used to run a proteomic analysis and reveal potentially relevant factors for drug resistance and tumor malignancy [67].

#### 3.2.2. Non-Endothelial Cells and Vessel-Associated Matrix

The increase in branching points and the heterogeneity of the vessels’ diameter are important changes in glioblastoma, which can be visualized and are directly related to changes in astrocytic end-feet arrangement. Typical imaging findings of glioblastoma include its infiltrative nature evident on T2-weighted or FLAIR images, heterogeneous signal intensity, and peripheral enhancement with regions of internal hemorrhage sparse in multifocal locations. The details of one morphological analysis showed a full-thickness multilayered fenestration involving the basement membrane and the astrocytic processes [68]. Whether these morphological features disrupt vascular integrity is still to be proven [69,70]. However, the immune cell margination and integration in the glioblastoma could be influenced by the different fenestration patterns from the endothelial lumen to the tumor core. T-cell invasion is not prominent but is evident in glioblastoma and positively correlates with tumor growth, being an exception to the paradigm of tumors thriving in an immune-free environment [69]. T-cells are consistently observable in a glioblastoma mass and show increased motility within the tumor [71]. The diapedesis of leukocytes, in general, is allowed by endothelial cells (CD31+ elements) preserving their adhesion properties. The endothelial fenestration exposes basal membrane proteins to the bloodstream, allowing a direct interaction between the integrins (i.e., VLA-1) on lymphocytes and collagen IV [72]. Glioblastoma-associated monocytes and macrophages, on the other hand, derive both from resident microglia and infiltrated monocytes. While the correlation between an increased proliferation of the tumor and the bulk of infiltrating immune cells is clear, the causative mechanism is still debated. 

With those premises, glioblastoma could be an ideal candidate for active immunotherapy, with a fenestrated and highly adhesive blood–tumor interface and with significant morpho-functional differences compared to the blood–brain barrier from the astrocytic end-feet through the basal lamina to the endothelial lumen. However, recent immunotherapy trials involving PD1/PD-L1 or CAR-T cells, immune checkpoint blockade, or vaccines failed to show significant benefits in glioblastoma patients [73,74].

The most accepted idea is that the high prevalence of immunosuppressive myeloid elements in the tumor core and the lack of active lymphocytes makes the glioblastoma “immune cold”, with a low response rate to immunotherapy [75]. According to this theory, underestimating myeloid cells’ activity on the infiltrating lymphocytes justifies clinical failures. A novel strategy could consider the reprogramming of microglia/macrophages counter-hijacking the glioblastoma survival strategy. To obtain this achievement, it is necessary to understand the subtle paracrine interplay between stromal, immune, and tumor cells, which is locally effective and highly dynamic. 

One work tried to establish a map of cellular spatial, molecular, and functional heterogeneity both with different low-grade glioma and newly diagnosed or recurrent glioblastoma [76]. These authors demonstrated the spatial diversity of immune infiltrates, glioma cells, and pericytes both among patients and within each patient. The authors provided S100A4 as a potential candidate for targeted immunotherapy. 

### 3.3. Third Pillar: CNS Invasion

Vascular cells and tumor-associated myeloid cells are the major constituents of a glioblastoma mass [77]. Myeloid cells were thought to be derived from resident microglia or marginating monocytes/macrophages, both accelerating glioblastoma cell invasion [78]. Instead, the rich vascular network is important for tumor–trophic functions, reshaping the CNS vasculature [77]. Microglia are distinct from bone-marrow-derived white cells but share their biomarkers and principal functions, from interfering with neuronal plasticity to innate immunity [79]. The classical boundary of invasion in the physiological CNS is considered the periventricular (choroid plexus) and perivascular zones, where the macrophages can be found, whereas other parts of the CNS are in the microglial domain [80]. The two cytotypes are abundant during pathological stimuli, and the peripheral immune cells can accumulate during chronic insults [79].

These notions are questioned while preserving the central role of tumor-associated myeloid cells in all the pillars we have described. Glioblastoma progression is well-defined in various models, but the ontology of different cytotypes and parenchymal progenitor cells is largely unknown. Using a transgenic lineage enabling the tracing of progenitor cells of the brain and the tumor, the authors of one study revealed a very intriguing subset of the cellular population derived from the glioblastoma itself and showing a myeloid expression profile [81]. These newly identified myeloid-like cells are traced back to a local progenitor and are not derived from the bone marrow. Therefore, this lineage, called tumor-associated cells with a myeloid-like expression profile (TAMEP), can be separated from microglia and macrophages and is generated by CNS-resident, SOX2-positive progenitors to enhance tumor growth and invasion, stimulating neo-angiogenesis.

To understand the properties and molecular pathways of human glioblastoma, a powerful mass spectrometry technique called SWATH (sequential window acquisition of all theoretical fragment ion) has been used. Pairing two independent cohorts of newly diagnosed and recurrent glioblastomas enabled authors to sort for recurrence-associated proteins. These proteins were further validated using immunohistochemistry and studied on xenografts and human organotypic brain slice cultures. The mechanisms were associated with transcriptomic data in both single-cell and bulk RNA sequencing. 

Three proteins have been identified as upregulated in recurrent glioblastoma and highly heterogeneous across patients: brain enriched myelin associated protein 1 (BCAS1), inverted formin 2 (INF2), and F-Box Protein 2 (FBXO2) [82]. 

In vitro, the knockout of these three genes did not affect the growth or clonogenicity assays but increased the migratory capacity of two cell lines: human glioma cell line LN-229 and the glioma-initiating cell line ZH-161. However, cell growth in vivo is more complex and depends on multiple interactions with the surrounding environment, considering cellular and non-cellular components, such as ECM, metabolites, hormones, and growth factors. The only orthotopic xenograft model that maintained reliable improved survival with the gene knockout was FBXO2. The animal model showed at the onset of neurological symptoms the presence of the tumor with reduced growth and invasion compared to the controls. The same tumoral line (LN-229) with or without FBXO2 knockout was injected into human organotypic brain slices to mimic the human brain microenvironment, and as expected, the control manifested a more infiltrative behavior. In contrast, LN-229 knockout cells exhibited a spotty growth pattern with a reduced affected area. However, knocking in the FBXO2 gene in a negative cell line model did not enhance CNS invasion. The infiltrative behavior is probably not inherited by FBXO2 expression but by concurrent associated cellular mechanisms. The expression levels of FBXO2 were studied within TCGA database and confronted with MRI imaging. The glioblastoma with high expression of FBXO2 presented reduced tumor sphericity compared to the highly-expressing tumors. This phenomenon matched the infiltrative pattern, also paired with a peculiar spatial transcriptomic expression of the gene [36,83]. FBXO2-positive glioma cells were mostly associated with regions of the tumor with high immune reactivity and neurodevelopmental reactivation of the transcription program, while BCAS1 and INF2 were less specific and more widely diffused across the different regions of the glioblastoma. The NPC- and OPC-like signatures were mostly associated with the FBXO2-enriched areas, confirming the strong pairing between the necrotic core and the infiltrative behavior. The spatial transcriptomic data in healthy tissue showed the expression of FBXO2 in the first layer of the neocortex (LI), where neural progenitors are harbored [84]. The gene is associated with both recurrence and infiltrative behavior of the glioblastoma; positive cells express pathways associated with synaptic plasticity, vesicular secretion, transmembrane transport, and cellular interactions with phospholipase C activation. The PSD-95 puncta were reduced in the areas surrounded by FBXO2-knockout cells, confirming in a complex in vitro model (brain slice cultures) the FBXO2 property of enhancing cellular interactions [82].

The knockout of FBXO2 favored the host survival in the orthotopic xenograft and reduced the invasion of organotypic brain slices in culture. The tumor infiltration zone was indeed enriched with FBXO2-positive elements. The function of FBX02 in physiology is a ubiquitin ligase substrate adaptor particularly expressed in the brain of patients with recurrent glioblastoma and less abundant in long-surviving patients.

Surgery remains the gold-standard procedure; the infiltrative glioblastoma tissue is distinguished from the healthy brain due to macroscopic features such as color, consistency, and vascularization. However, significant heterogeneity among patients is possible. These tumors have different appearances depending on the cellularity and molecular composition, varying from grayish and reddish to yellowish; they may or may not have a cystic/necrotic core, and their consistency can range from fibrous to gelatinous. There is pronounced neovascularization, which disrupts the normal pattern of vessel distribution and NVU permeability. Moreover, thrombosed veins are a characteristic feature within glioblastoma [85]. From a neuroradiological standpoint, MRI remains the gold standard for visualizing glioblastoma. On T1-weighted sequences, post-gadolinium injection, an enhancing nodule (EN) with a necrotic component is often discerned. T2-weighted sequences, on the other hand, highlight the water content in the cerebral parenchyma. Specifically, the fluid-attenuated inversion recovery (FLAIR) T2w images emphasize pathological processes by suppressing the T2 signal from cerebrospinal fluid (CSF), revealing the “peritumoral area” that consists of infiltrated white matter [86,87]. Histologically, FLAIR areas consist of fragments of white matter, either focally or diffusely infiltrated by tumor cells (76% of cases), a mix of white matter with reactive astrogliosis and grey matter with perineuronal satellitosis (15%), and tumor tissue (9%) [88]. Iang et al. retrospectively analyzed 585 patients, classifying them based on the Volume FLAIR–Volume EN ratio. Those with a Volume FLAIR–Volume EN ratio <10 were defined as having a “proliferation-dominant” subtype growth pattern, while glioblastomas with a ratio >10 were termed “diffusion-dominant” subtypes [89]. It is suggested that the Gompertzian growth is the most reliable mathematical model to predict tumor growth [90]. In our previous study, we first described the mean Velocity of Diameter Expansion (VDE) as 39.9 mm/year and 45.2 mm/year for FLAIR volume and EN, respectively. The VDE analysis showed a peak in variation over time. The growth pattern and evolution are still studied through the classical histological features, including pluricellularity, the formation of hypoxic pseudopalisades, aberrant mitoses, and glomeruloid vessels. The cerebral blood volumes positively correlate with hypoxia and EN in highly infiltrating glioblastoma, supporting the strict association among hypoxia/necrotic core, angiogenesis, and CNS invasion [91]. Many authors underline the correlation between spatial locations and biological features and suggest that the temporal peri-atrial white matter location predicts high migration potential and poor prognosis independently of other known prognostic clinical variables [23,92]. Other groups, moreover, demonstrated the possibility of predicting the mutation of IDH1 and PTEN and the methylation of MGMT using radiological data, proposing a probabilistic neuroradiological atlas of different phenotypic glioblastomas [4,93,94,95,96,97]. 

The migration potential can be understood by studying the glioblastoma cellular population of the peripheral zones of the mass. These areas are mainly composed of glioma-associated microglia and macrophages [78,98] or may hide in plain sight an ontologically different myeloid cell, as demonstrated with TAMEP. 

Tumor-associated myeloid cells infiltrate high-grade gliomas and orchestrate the immune response, and the prognostic value of different myeloid cluster-specific gene sets was evaluated in TCGA [99]. The profiling and unbiased clustering of 24,227 myeloid cells from glioblastoma and high-grade astrocytoma with IDH mutation identified nine myeloid cell clusters, including monocytes, six tumor-associated microglia/macrophage subsets, and dendritic cells. The different clusters manifest transcriptional diversity controlled by peculiar regulons. These subsets show spatial enrichment of distinct clusters at peri-vascular and necrotic areas, differing at the tumor–brain interface, corroborating the role of these cells in the glioblastoma invasion of the CNS and the mesenchymal transition [99]. 

The immune escape or coldness of glioblastoma, a key to the invasion of the surrounding brain tissue, has been studied by applying an in silico multidimensional model with spatially resolved and single-cell gene expression data of 45,615 immune cells from 12 tumor samples [100]. The study revealed a subset of myeloid cells, spatially localizing to mesenchymal-like tumor regions. These cells are HMOX1-positive and release IL-10, driving T-cell exhaustion and thus reducing the immune reactivity of the microenvironment [100]. These in silico findings have been validated using a human cortical glioblastoma model inoculated with patient-derived peripheral T-cells for the immune compartment. This model recapitulates the immunological dysfunctions and the transformation of T-cells. The inhibition of the JAK/STAT pathway rescued in vitro the T-cell functionality and was confirmed in an animal model [100]. 

The frequency of recurrence of glioblastoma could also be related to the CSCs. Diverse studies found that CD133 expression correlated with the pattern of recurrence [101,102]. CD133 expression rate differences were evident among MRI-distinct groups of patients, with significantly higher levels in the SVZ. The central role of the CNS anatomy plays along with the molecular characteristics of the CSCs to predict the location of the recurrence of glioblastoma. Indeed, astrocyte-like CSCs with high CD133 expression levels were reported to leave the SVZ and migrate into distant brain regions [103].

A recent study was not able to replicate this finding [104]. However, the authors demonstrated that the combined analyses of CD133 expression and tumor localization can be used to predict the pattern of recurrence in glioblastoma. Indeed, 70% of high CD133 expression (≥15%) tumors showed distant recurrence patterns. The glioblastoma masses with low or no expression of CD133 could develop adhesive properties and express CD44 as a stemness marker [105]. Refining molecular markers and anatomical features is promising in predicting proliferative or migration properties in glioblastoma.

## 4. Conclusions

The growth patterns and evolution of glioblastoma are still unclear, even considering the big data and spatial multi-omics approaches standing alongside the design of new clinical trials. Classical histological features are fundamental for understanding the relationship between the cellularity and vascularization of the tumor, which varies even within the same patient (Figure 3). This review stresses the correlation between anatomical, biological, and fundamental features of this tumor type, without reducing its complexity but suggesting that the necrotic core, the vascular compartment, and the CNS invasion need to have a common denominator to be found to solve the enigma.

## Figures and Tables

**Figure 1 cells-13-01754-f001:**
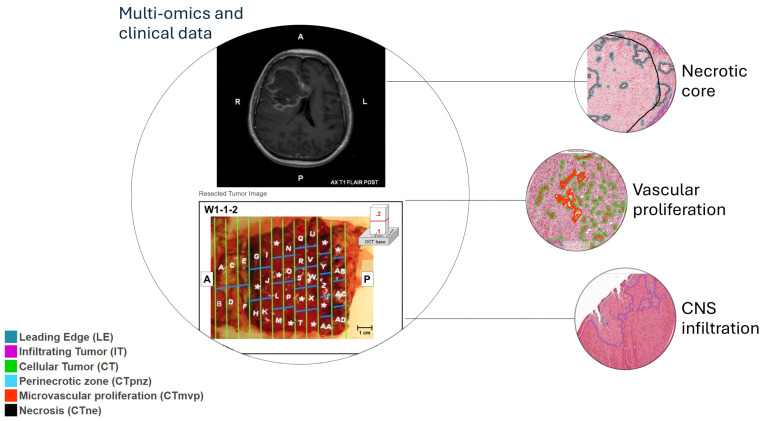
The three pillars of glioblastoma multiforme. Descriptive images of microscopic characteristics of each pillar within macroscopic radiological and surgical specimens of patients are accessible from the public multi-omics dataset Ivy Glioblastoma Atlas Project (https://glioblastoma.alleninstitute.org/ish, accessed on 25 August 2024). The radiological and surgical specimens are oriented with letters and symbols, that are used for navigating the tool.

**Figure 2 cells-13-01754-f002:**
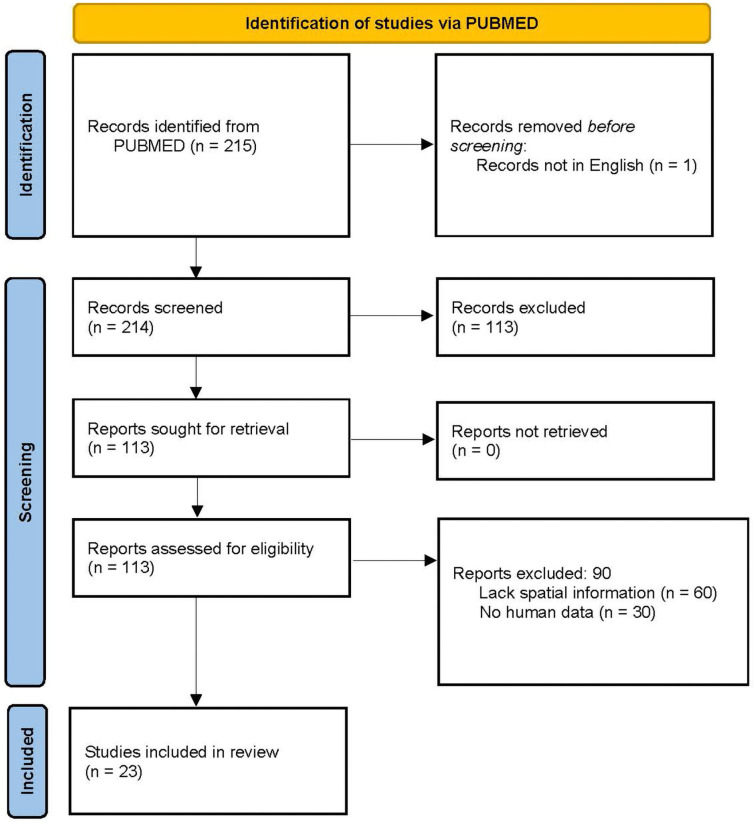
PRISMA flow diagram of the systematic review [30]. This work is licensed under CC BY 4.0. To view a copy of this license, visit https://creativecommons.org/licenses/by/4.0/.

**Figure 3 cells-13-01754-f003:**
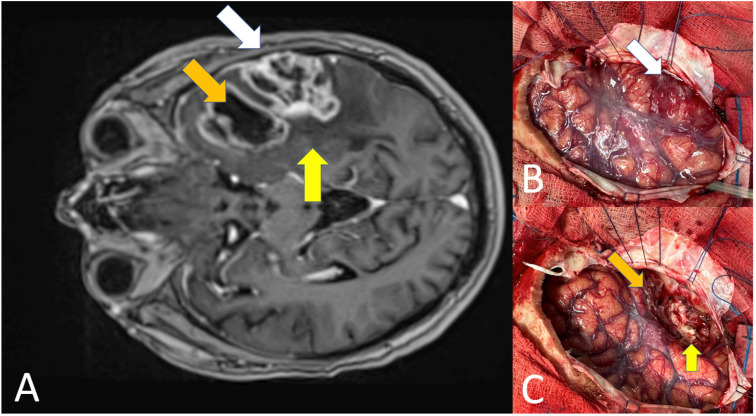
The images show the three pillars of glioblastoma in an axial section of an MRI image T1-weighted and enhanced with gadolinium (**A**) correlating it with intraoperative findings before excision (**B**) and after initial resection of the enhancing nodule (**C**). 1. necrotic core (orange arrow); 2. vascular proliferation (white arrow); 3. CNS infiltration (yellow arrow).

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
