# Peer review of "The Three Pillars of Glioblastoma: A Systematic Review and Novel Analysis of Multi-Omics and Clinical Data"

_cells, 2024, doi:10.3390/cells13211754_

Round 1

Reviewer 1 Report

Comments and Suggestions for Authors

The paper by De Luca et al. investigate the role of necrotic core, vascular proliferation and parenchyma infiltration as major determinants  of GBM outcome. Authors analyzed  multi-omics and clinical data to draw a picture of tumor inter- and intra-tumor heterogeneity . The study resumes potentially interesting findings, the analysis did not completely technically sounds and conclusions are relatively supported by literaturei. The manuscript is quite clearly written.

Some points need to be clarified:

1.       In Materials & Methods, it is not clear how results were analyzed and processed by  ZOTERO, since selection of articles can be made with filters in PubMed.

2.       It should be useful detailing why the 23 selected paper from 2021 to 2024 (for an homogeneous classification of GBM in the last WHO 2021) for the review are probably included in a larger number of older papers, likely differing in classification criteria.

3.       Glioblastomas genetic features (IDH1/2-wild-type, TERT promoter mutation, EGFR amplification, and/or combined whole chromosome 7+/10- chromosome ) playing a relevant role in GBM biology should be at least cited among determinant factors in its development and recurrence, as a potential common features impacting o the 3 pillars.

4.       The exclusion of the high mitotic activity of GBM cells from  hallmarks of GBM should be explained , although microvascular proliferation, necrosis and infiltration are main drivers of its aggressiveness.

5.       Reference n. 40 lack of year of publication.

Comments on the Quality of English Language

The manuscript is quite clearly written.

Author Response

  1. In Materials & Methods, it is not clear how results were analyzed and processed by ZOTERO, since selection of articles can be made with filters in PubMed.2

The statement could be misleading so we modified the sentence: Results were stored and managed with the ZOTERO reference manager (Center for History and New Media, George Mason University, Virginia, VA, USA)”

  1. It should be useful detailing why the 23 selected paper from 2021 to 2024 (for an homogeneous classification of GBM in the last WHO 2021) for the review are probably included in a larger number of older papers, likely differing in classification criteria.

We modified the Materials and Methos section to make it more detailed and explicitly cite the bias of a discussion with papers possessing mixed classification criteria.

  1. Glioblastomas genetic features (IDH1/2-wild-type, TERT promoter mutation, EGFR amplification, and/or combined whole chromosome 7+/10- chromosome ) playing a relevant role in GBM biology should be at least cited among determinant factors in its development and recurrence, as a potential common features impacting o the 3 pillars.

We wrote a paragraph including the key features of glioblastoma molecular biomarkers and how these can specifically impact intratumoral heterogeneity.

  1. The exclusion of the high mitotic activity of GBM cells from hallmarks of GBM should be explained , although microvascular proliferation, necrosis and infiltration are main drivers of its aggressiveness.

We agree with the reviewer that the mitotic activity of glioblastoma is a key feature, however, we explained in the introduction that we think “ […] the current understanding of glioblastoma as an uncontrolled proliferating tissue can be discarded. The glioblastoma develops within the central nervous system (CNS) and perfectly adapts to its features, mimicking CNS components' anatomy and metabolism. The glioblastoma is a system fully equipped to hack the CNS and therefore cannot metastasize elsewhere […]”. Moreover, the high mitotic activity can be considered the common element of the three pillars, accounting for clonal selection in the necrotic core, metabolic demand for microvascular proliferation, and infiltrative potential, as proliferating positive elements are usually present along the migratory pathways of glioblastoma. We included this statement in the manuscript.

  1. Reference n. 40 lack of year of publication.

We updated the reference with the year of publication

Reviewer 2 Report

Comments and Suggestions for Authors

The review gives a comprehensive depiction of GBM-related characteristics by discussing the three Pillars that support cancer and its potential therapy resistance.

As a review, I recommend including references, particularly in sections where there are no bibliographic references.

The guidelines for GBM classification have recently been updated. I request that the authors better define the tumor context in which they describe some parameters, specifically in terms of tumor heterogeneity.

Comments on the Quality of English Language

Broken English, some terms appear out of context and misleading.

Author Response

We thank the reviewer for the suggestions and extensively checked the manuscript to improve the English writing. We included 14 additional references to the bibliography of the manuscript. We included details about the novel WHO classification and specifically focused on tumor heterogeneity.

Reviewer 3 Report

Comments and Suggestions for Authors

In this manuscript, De Luca and colleagues presented the three pillars of glioblastoma that greatly support the survival of this type of cancer cells, namely divergent molecular pathways and anatomical features. Overall, the review is of great interest and the topics are up-to-date and well-written. Only few issues were detected by this reviewer, that should be addressed by the authors before publication. Therefore, this reviewer considers the paper is suitable for publication in Cells upon minor revision.

- According to the most recent WHO classification of CNS tumours, the term glioblastoma multiforme is misused. Therefore, I would recommend the authors to use glioblastoma only. In line with this, there is no need to use an acronym for this word.

- Throughout the manuscript, there are some Latin expressions not written in italic. Please give attention to this.

- In the abstract (lines 30-31), the words cancer stem cells and neurovascular unit only are referred once, so there is no need to define their acronyms in this section.

- Lines 35-37: I would suggest to change this to “Glioblastoma is the most frequent and fatal primary malignant brain tumor, which median survival for patients receiving the current evidence-based treatment is between 12 and 15 months [1, 2].”.

- Lines 47-48: There is no need to divide this info into two paragraphs.

- Lines 54-57: This sentence is confusing, please rewrite this to make it more understandable.

- Line 82: I would suggest to change “anatomic” to “anatomical”.

- Line 83: Please correct this sentence to “Our data also verify the hypothesis that different tumors arising from definite areas and diverse spatial locations are associated with specific features [15].”.

- Please correct the legend for Figure 1 to “Descriptive images of microscopic characteristics of each pillar within macroscopic radiological and surgical specimens of patients are accessible from the public multi-omics dataset Ivy Glioblastoma Atlas Project (https://glioblastoma.al-107 leninstitute.org/ish).”.

- Materials and methods section needs to be described in more detail or rewritten to be more understandable. It might be interesting if the first paragraph of the results (lines 118-123) and Figure 2 where allocated to materials and methods section, since there are details about exclusion criteria.

- Lines 133-14: please correct this to “Based on the signature of these cells, the classification generated two subtypes via consensus clustering.”.

- Lines 190-191: this sentence is confusing. Please rewrite it.

- Lines 222-227: I would suggest to change this to “These therapies, and other antitumor compounds tested, including low molecular weight compounds, must endeavor against the blood-brain interface, characterized by a well-defined unit from the vascular lumen to the synaptic cleft, the NVU, granting the homeostasis of the CNS. However, failure to cross the first space of the unit, the so-called blood-brain barrier (BBB), is one of the unique features of the brain that has led to negative outcomes of tested drugs.”.

- Lines 239-242: Please rewrite this to make it more understandable.

- Lines 265-269: Please rewrite this to make it more understandable.

- Lines 375-376: I would suggest the author to describe which two cell lines are they referring to.

Author Response

- According to the most recent WHO classification of CNS tumours, the term glioblastoma multiforme is misused. Therefore, I would recommend the authors to use glioblastoma only. In line with this, there is no need to use an acronym for this word.

We checked the manuscript and revised the term as suggested.

- Throughout the manuscript, there are some Latin expressions not written in italic. Please give attention to this.

We carefully checked the manuscript for Latin terms and expressions to write them in Italic

- In the abstract (lines 30-31), the words cancer stem cells and neurovascular unit only are referred once, so there is no need to define their acronyms in this section.

We updated the abstract session, based on the suggestions of this reviewer and the Editor.

- Lines 35-37: I would suggest to change this to “Glioblastoma is the most frequent and fatal primary malignant brain tumor, which median survival for patients receiving the current evidence-based treatment is between 12 and 15 months [1, 2].”.

We agree with this reviewer's suggestion.

- Lines 47-48: There is no need to divide this info into two paragraphs.

We removed the paragraph.

- Lines 54-57: This sentence is confusing, please rewrite this to make it more understandable.

We rephrased the Lines 54-57 to make them more understandable

- Line 82: I would suggest to change “anatomic” to “anatomical”.

The words “anatomic” and “anatomical” have been swapped.

- Line 83: Please correct this sentence to “Our data also verify the hypothesis that different tumors arising from definite areas and diverse spatial locations are associated with specific features [15].”

The sentence has been rephrased.

- Please correct the legend for Figure 1 to “Descriptive images of microscopic characteristics of each pillar within macroscopic radiological and surgical specimens of patients are accessible from the public multi-omics dataset Ivy Glioblastoma Atlas Project (https://glioblastoma.al-107 leninstitute.org/ish).”

We updated the legend for Fig. 1.

- Materials and methods section needs to be described in more detail or rewritten to be more understandable. It might be interesting if the first paragraph of the results (lines 118-123) and Figure 2 where allocated to materials and methods section, since there are details about exclusion criteria.

The Materials and methods section has been rewritten to include more details, including the reference to Figure 2.

- Lines 133-14: please correct this to “Based on the signature of these cells, the classification generated two subtypes via consensus clustering.”

The sentence has been corrected.

- Lines 190-191: this sentence is confusing. Please rewrite it.

The sentence has been rewritten as: “The tumor stroma is filled with tightly packed macrophages/microglia whose role in contributing to a poor prognosis needs to be elucidated

- Lines 222-227: I would suggest to change this to “These therapies, and other antitumor compounds tested, including low molecular weight compounds, must endeavor against the blood-brain interface, characterized by a well-defined unit from the vascular lumen to the synaptic cleft, the NVU, granting the homeostasis of the CNS. However, failure to cross the first space of the unit, the so-called blood-brain barrier (BBB), is one of the unique features of the brain that has led to negative outcomes of tested drugs.”.

The sentence has been corrected.

- Lines 239-242: Please rewrite this to make it more understandable.

These lines have been changed into: “Endothelial cells regulate the passage of water-soluble molecules independently of their molecular weight through pinocytosis and vesicular transcytosis. All these characteristics are important for CNS homeostasis, and data from cellular transcription isolated from abnormal vessels in glioblastoma show a distinct gene signature and vascular proliferation as key features among all glioblastoma phenotypes [37–39].

- Lines 265-269: Please rewrite this to make it more understandable.

These lines have been rephrased as: “Pericytes mediate neoangiogenesis in the glioblastoma mass, contacting both tumoral perivascular cells and brain endothelium through Semaphorin-4D (SEMA4D) and focal adhesion kinase (FAK). The SEMA4D protein or Cluster of Differentiation 100 (CD100), is a well-known cell-guidance factor that seems crucial for pericyte migration and neoangiogenesis in combination with high levels of FAK expression. SEMA4D and FAK are associated with poorer prognosis in high-grade gliomas”

Lines 375-376: I would suggest the author to describe which two cell lines are they referring to.

The cell lines have been specified.